# Sensory Reactivity Phenotype in Phelan–McDermid Syndrome Is Distinct from Idiopathic ASD

**DOI:** 10.3390/genes12070977

**Published:** 2021-06-26

**Authors:** Teresa Tavassoli, Christina Layton, Tess Levy, Mikaela Rowe, Julia George-Jones, Jessica Zweifach, Stacey Lurie, Joseph D. Buxbaum, Alexander Kolevzon, Paige M. Siper

**Affiliations:** 1School of Psychology and Clinical Language Sciences, University of Reading, Berkshire RG6 6BZ, UK; teresa.tavassoli@gmail.com; 2Seaver Autism Center for Research and Treatment, Icahn School of Medicine at Mount Sinai, New York, NY 10029, USA; christina.layton@mssm.edu (C.L.); tess.levy@mssm.edu (T.L.); jessica.zweifach@mssm.edu (J.Z.); joseph.buxbaum@mssm.edu (J.D.B.); alexander.kolevzon@mssm.edu (A.K.); 3Department of Psychiatry, Icahn School of Medicine at Mount Sinai, New York, NY 10029, USA; 4Radiology and Biomedical Imaging, University of California San Francisco, San Francisco, CA 94143, USA; rowem513@gmail.com; 5School of Psychology, University of Texas at Austin, Austin, TX 78712, USA; jgeorgejones@utexas.edu; 6Ferkauf Graduate School of Psychology, Yeshiva University, Bronx, NY 10461, USA; staceymlurie@gmail.com; 7Mindich Child Health and Development Institute, Icahn School of Medicine at Mount Sinai, New York, NY 10029, USA; 8Friedman Brain Institute, Icahn School of Medicine at Mount Sinai, New York, NY 10029, USA; 9Department of Genetics and Genomic Sciences, Icahn School of Medicine at Mount Sinai, New York, NY 10029, USA; 10Department of Neuroscience, Icahn School of Medicine at Mount Sinai, New York, NY 10029, USA; 11Department of Pediatrics, Icahn School of Medicine at Mount Sinai, New York, NY 10029, USA

**Keywords:** Phelan–McDermid syndrome, autism spectrum disorder, sensory reactivity

## Abstract

Phelan–McDermid syndrome (PMS) is one of the most common genetic forms of autism spectrum disorder (ASD). While sensory reactivity symptoms are widely reported in idiopathic ASD (iASD), few studies have examined sensory symptoms in PMS. The current study delineates the sensory reactivity phenotype and examines genotype–phenotype interactions in a large sample of children with PMS. Sensory reactivity was measured in a group of 52 children with PMS, 132 children with iASD, and 54 typically developing (TD) children using the Sensory Assessment for Neurodevelopmental Disorders (SAND). The SAND is a clinician-administered observation and corresponding caregiver interview that captures sensory symptoms based on the DSM-5 criteria for ASD. Children with PMS demonstrated significantly greater hyporeactivity symptoms and fewer hyperreactivity and seeking symptoms compared to children with iASD and TD controls. There were no differences between those with Class I deletions or sequence variants and those with larger Class II deletions, suggesting that haploinsufficiency of *SHANK3* is the main driver of the sensory phenotype seen in PMS. The syndrome-specific sensory phenotype identified in this study is distinct from other monogenic forms of ASD and offers insight into the potential role of *SHANK3* deficiency in sensory reactivity. Understanding sensory reactivity abnormalities in PMS, in the context of known glutamatergic dysregulation, may inform future clinical trials in the syndrome.

## 1. Introduction

Autism spectrum disorder (ASD) is characterized by difficulties in social communication and repetitive behaviors [1] with a prevalence of approximately 1 in 54 children [2]. Known genetic causes of autism now account for 30% of cases [3,4,5]. Phelan–McDermid syndrome (PMS) is one of the most common genetic forms of autism and is present in up to 2% of affected cases. PMS is caused by microdeletions in the long arm of chromosome 22 which includes the *SHANK3* gene or by pathogenic variants in *SHANK3* [6,7]. *SHANK3* encodes a scaffolding protein in glutamate synapses, with animal models indicating glutamatergic dysregulation in PMS [8,9]. Individuals with PMS have a wide range of symptoms including intellectual disability (ID), delayed or absent speech, hypotonia, medical comorbidities, and dysmorphic features [10]. Up to 80% of children with PMS also meet Diagnostic and Statistical Manual of Mental Disorders (DSM) criteria for ASD [11]. As defined by the DSM-5, sensory reactivity symptoms fall within the Restricted and Repetitive Behavior domain and include hyperreactivity (over-responsiveness), hyporeactivity (under-responsiveness), and seeking out sensory aspects of the environment (seeking) [1]. Hyperreactivity describes a strong and/or aversive reaction to a sensory stimulus, such as covering ears in response to everyday sounds or avoiding wearing clothes of a certain texture. Hyporeactivity describes delayed or absent responses to sensory stimuli such as failure to notice the sound of an alarm or the sight of a car passing by. Sensory seeking refers to a craving and fascination with certain sensory stimuli, such as repeatedly touching specific textures or visually inspecting objects for extended periods of time.

While sensory reactivity symptoms are widely reported in idiopathic forms of ASD (iASD) [12,13] with sensory reactivity differences reported in approximately 60-85% of autistic children [12,14,15], few studies have examined sensory symptoms in PMS [16]. Results from our group using the Short Sensory Profile, a well-validated caregiver questionnaire, found that children with PMS showed more hyporeactivity symptoms, specifically low energy/weak symptoms, and less hyperreactivity, specifically auditory/visual sensory sensitivity, as compared to children with iASD [16]. In addition, high levels of sensory hyporeactivity were identified in individuals with sequence variants in *SHANK3*, with pain insensitivity reported in 94% of cases. Deep characterization of the sensory reactivity phenotype in PMS holds value for the clinical evaluation and treatment of individuals with PMS and for a better understanding of potential relationships with underlying biological mechanisms related to the *SHANK3* gene. More broadly, robust methods to quantify sensory reactivity symptomatology are especially important for individuals who are severely affected, including individuals with PMS, who are often unable to verbally describe their sensory experiences. 

From a biological perspective, SHANK3 is known to affect glutamatergic processing. However, the relationship between the potential role of glutamate and sensory phenotypes remains unclear. There are some reports linking GABAergic activity to sensory reactivity. For example, in typically developing children, specific polymorphisms in the *GABRB3* gene were significantly associated with tactile reactivity [17]. There is also some evidence of an association between heterozygous *Gabrb3* deletion and increased tactile reactivity in male mice [18]. Overall, however, little is known about the role of glutamatergic/GABAergic neurotransmission in the emergence and longer term trajectory of sensory symptoms. Understanding sensory reactivity in a syndrome caused by glutamatergic dysfunction may offer critical information about the relationship between clinical features and underlying neurobiology.

The current study delineates the sensory reactivity phenotype and examines genotype–phenotype interactions in a large sample of children with PMS. The Sensory Assessment for Neurodevelopmental Disorders (SAND) [19] was used to quantify sensory symptoms and examine its utility as a clinical outcome assessment. The SAND is a clinician-administered observation and corresponding caregiver interview that captures sensory symptoms based on DSM-5 criteria for ASD and is appropriate for individuals with varying levels of ability, including those with few to no words [20]. The SAND has been used successfully in other genetic forms of autism [21]; however, it has yet to be examined extensively in PMS. Establishing the utility of the SAND as a clinical outcome measure may have important implications for ongoing clinical trials in PMS.

## 2. Materials and Methods

### 2.1. Participants

Sensory reactivity was measured in 52 children with PMS (26 males, *M* age = 6.65, SD = 2.90), 132 children with iASD (114 males, *M* age = 6.11, *SD* = 2.55), and 54 typically developing (TD) children (24 males, *M* age = 5.39, SD = 2.55), between the ages of 18 months and 12 years old (Table 1). A subset of children with PMS (*n* = 17) and iASD (*n* = 30) returned for a 12-week visit to assess the stability of the SAND during a typical clinical trial interval. PMS diagnosis was confirmed using chromosomal microarray or sequencing [11]. ASD was diagnosed according to a consensus diagnosis determined by psychiatric evaluation using the DSM-5, the Autism Diagnostic Observation Schedule, Second Edition [22], and the Autism Diagnostic Interview-Revised [23]. TD participants were screened with the Social Responsiveness Scale, Second Edition, and all scored below clinically significant cutoffs [24]. Appropriate cognitive assessments were determined based on age and language ability and included the Mullen Scales of Early Learning [25], the Stanford–Binet, Fifth Edition (SB-5) [26], or the Differential Ability Scales, Second Edition (DAS-2) [27]. Groups differed significantly in age (*p* = 0.048) and FSIQ/DQ (*p* < 0.001). Given the level of intellectual disability common in children with PMS, TD participants could not be matched on IQ; however, a sample of children with iASD also met the criteria for intellectual disability (*n* = 55, mean FSIQ = 44.57, SD = 12.97, range 20–67). Analyses were run with and without IQ and age as covariates. Moreover, children with iASD and PMS with an IQ below 70 were also compared directly.

### 2.2. Ethics Declaration

The study was approved by the Mount Sinai Program for the Protection of Human Subjects (Study: 98-0436, Assessment Core for phenotyping approved annually since 1998). Informed written consent was obtained from parents or legal guardians and assent was obtained from participants when appropriate.

### 2.3. Sensory Evaluation

The Sensory Assessment for Neurodevelopmental Disorders (SAND) [20] is a standardized clinician-administered observation and corresponding caregiver interview validated in children ages 2–12. Children are presented with a series of visual (e.g., spinning disc, light up wand), tactile (e.g., textured balls, hot/cold packs), and auditory stimuli (e.g., unexpected noisemaker, siren buzzer) that probe for sensory behaviors, which are rated by a trained examiner on an algorithm measuring discrete sensory hyperreactivity, hyporeactivity, and seeking behaviors across visual, tactile, and auditory modalities. The corresponding caregiver interview consists of 36 corresponding items and indicates whether a given sensory behavior is present (1) or absent (0). For any domain with behaviors coded “present,” clinicians/caregivers rated the severity of symptoms within that domain: mild (1) or moderate-to-severe (2). Total SAND scores are based on the combination of observed and reported behaviors, including severity scores. Summary scores were calculated for each DSM-5 symptom domain (hyperreactivity, hyporeactivity, seeking), modality (visual, tactile, auditory), and subscale (e.g., visual hyperreactivity, visual hyporeactivity, visual seeking, etc.). Total scores greater than or equal to 16, domain and modality scores greater than or equal to 8, and subscale scores greater than or equal to 5 reflect a clinically significant level of symptoms. 

### 2.4. Analysis

SPSS 27 was used to analyze the data. Descriptive statistics examined the percentage of participants who fell above clinical cutoffs on the SAND. A MANOVA was conducted to test if groups differed by domain (hyperreactivity, hyporeactivity and seeking) and modality (visual, tactile, auditory). A MANCOVA with age and IQ as covariates was run to see if age or IQ had an effect, and a MANOVA was re-run to examine the group with ASD + intellectual disability relative to the PMS group. Tukey multiple comparison corrections were used for all post hoc testing. Finally, to assess stability of SAND scores over a 12-week period, intraclass correlation coefficients (ICC) were calculated.

For genotype–phenotype analyses, participants with PMS were split into two groups based on genotype: Class I (*n* = 30) included participants with sequence variants in *SHANK3*, or those with deletions including only *SHANK3* or *SHANK3* with ACR, RABL2B, and/or ARSA; Class II (*n* = 22) included all other participants with larger deletions that did not classify for Class I. Mann–Whitney U tests were used to assess differences between groups in SAND scores. Chi-square tests assessed differences in the proportion that surpassed SAND clinical cut-off scores. Effect sizes were measured with Cohen’s d or phi. 

## 3. Results

### 3.1. Group Comparisons on SAND Domain, Modality and Subscale Scores

A MANOVA (Table 2) showed that groups significantly differed on SAND Total Score (F (18) = 28.53, *p* < 0.0001). Tests of between-subjects effects showed significant differences in each DSM-5 domain: hyperreactivity (F(2) = 45.85, *p* < 0.0001), hyporeactivity (F (2) = 114.89, *p* < 0.0001), and seeking (F(2) = 89.07, *p* < 0.0001). All sensory modalities also significantly differed: visual (F (2) = 81.21, *p* < 0.0001), tactile (F(2) = 93.46, *p* < 0.0001), and auditory (F(2) = 71.70, *p* < 0.0001). Differences between groups were significant for all subscales: visual hyperreactivity (F = 10.75, *p* < 0.0001), visual hyporeactivity (F (2) = 74.45, *p* < 0.0001), visual seeking (F(2) = 85.91, *p* < 0.0001), tactile hyperreactivity (F = 16.70, *p* < 0.0001), tactile hyporeactivity (F(2) = 71.67, *p* < 0.0001), tactile seeking (F(2) = 47.79, *p* < 0.0001), auditory hyperreactivity (F(2) = 40.37, *p* < 0.0001), auditory hyporeactivity (F(2) = 68.50, *p* < 0.0001), and auditory seeking (F(2) = 31.75, *p* < 0.0001).

Post hoc comparisons using Tukey multiple comparison corrections revealed that children in the iASD group significantly differed from children in the TD group in all domains, *p* < 0.0001 (hyperreactivity, hyporeactivity and seeking), modalities, *p* < 0.0001 (visual, tactile and auditory), and subscales, *p* < 0.0001 (Table 2).

The PMS and TD groups differed significantly in hyporeactivity (*p* < 0.0001) and seeking (*p* < 0.0001) domains across all three modalities (*p’s* < 0.0001). PMS and TD groups did not differ on hyperreactivity total scores (*p* = 0.28) or any hyperreactivity subscales. The PMS and TD groups differed significantly on the auditory (*p* < 0.0001) and tactile (*p* < 0.0001) seeking subscales, but not on the visual seeking subscale (*p* = 0.09) (Table 2). Repetitive non-communicative sounds (auditory) and mouthing of objects (tactile) represented two of the most common seeking behaviors observed in the PMS group.

Children with PMS and iASD significantly differed on all three domains (*p* < 0.0001) but not on any modality: visual (*p* = 0.13), tactile (*p* = 0.61), auditory (*p* = 0.98). Children with PMS and iASD differed significantly on visual hyporeactivity *(p* < 0.0001), visual hyperreactivity (*p* = 0.045), and visual seeking *(p* < 0.0001) subscales. There were also significant differences for tactile hyporeactivity (*p* < 0.0001) and tactile hyperreactivity (*p* < 0.0001) but not for tactile seeking (*p* = 0.16). There were significant differences on each auditory subscale: hyperreactivity (*p* < 0.0001), hyporeactivity (*p* < 0.0001), and seeking (*p* < 0.0001) (Figure 1 and Table 2).

When examining the percentage of participants who fell in the clinically significant range (Table 3), the PMS and iASD groups showed similar levels of sensory symptoms overall; however, symptoms in the PMS group were largely driven by hyporeactivity (e.g., delayed or absent responses to the sight, sound, and feel of stimuli) and seeking (e.g., mouthing of objects, making repetitive sounds). In the iASD group, high rates of seeking were identified along with relatively equal rates of hyperreactivity and hyporeactivity, likely reflecting the heterogeneity of ASD. With regard to modality, tactile symptoms were most common for both groups, followed by auditory and visual symptoms. There were no significant differences in sensory reactivity scores between males and females with PMS.

### 3.2. SAND Score Comparisons for Participants with Intellectual Disability

Group comparisons were conducted with children with iASD who also met DSM-5 criteria for ID based on IQ and Vineland scores <70 and compared to children with PMS, who all met the criteria for ID. Similar to comparisons described above, children with PMS showed significantly less hyperreactivity (*p* < 0.0001) and seeking (*p* < 0.0001), and more hyporeactivity (*p* < 0.0001). Children with PMS also showed fewer visual (*p* = 0.001) and auditory (*p* = 0.006) symptoms overall and no difference in the number of tactile symptoms (*p* = 0.22). Specifically, children with PMS had fewer visual hyperreactivity symptoms (*p* = 0.013), greater visual hyporeactivity symptoms (*p* < 0.0001), and fewer visual seeking symptoms (*p* < 0.0001) compared to the iASD + ID group. Children with PMS also showed significantly fewer tactile hyperreactivity symptoms (*p* = 0.03), greater tactile hyporeactivity symptoms (*p* < 0.0001), and less tactile seeking symptoms (*p* = 0.001). Children with PMS and iASD + ID significantly differed on all auditory subscales: hyperreactivity (*p* < 0.0001), hyporeactivity (*p* < 0.0001), and seeking (*p* < 0.0001).

### 3.3. Stability of SAND Scores

ICCs assessed absolute agreement in SAND scores for the PMS (*n* = 17) and iASD (*n* = 30) groups (Table 3) and indicated moderate to high levels of consistency over a 12-week period (Table 4).

### 3.4. Genotype–Phenotype Associations

The Class I group included 30 participants (7.3 year ± 2.8, 37% female) and the Class II group included 22 participants (5.7 year ± 2.8, 68% female). The analyses revealed that the Class I and Class II groups only differed on the SAND auditory subscales, where the Class I group had significantly higher hyperreactivity (*p* = 0.011, *d* = 0.89) and lower hyporeactivity (*p* = 0.02, *d* = 0.72) symptoms than the Class II group (Table 5). Similarly, the Class I group was significantly more likely to surpass the clinical threshold for auditory hyperreactivity (*p* = 0.044, phi = 0.279) and less likely for auditory hyporeactivity (*p* = 0.033, phi = 0.296) than the Class II group. The groups did not differ on any other SAND scales, and the results from the Mann–Whitney U tests did not undergo correction for multiple comparisons.

## 4. Discussion

This study demonstrated the utility of a direct observational assessment and corresponding caregiver interview to characterize sensory reactivity symptoms in PMS. Our results indicate that children with PMS not only differ from TD children, but also from children with iASD. Specifically, children with PMS showed more hyporeactivity symptoms and fewer hyperreactivity and seeking symptoms compared to children with iASD. Differences were particularly prominent in the auditory domain. The results were consistent when comparing individuals with PMS to children with iASD who also had ID. While both children with PMS and iASD had higher overall scores on the SAND, on average, compared to TD children, the PMS and TD groups did not differ in the frequency of hyperreactivity. Findings were consistent within the PMS group regardless of sex. 

This unique profile, in which children with PMS showed significantly more hyporeactivity symptoms and less hyperreactivity across visual, tactile, and auditory sensory modalities, suggests there may be a link between glutamatergic processing and sensory hyporeactivity. Results are similar to Mieses et al. (2016) [15] where we also found a significant number of hyporeactivity symptoms, specifically on the SSP low energy/weak scale, and fewer auditory hyperreactivity and auditory filtering difficulties in children with PMS compared to an independent iASD group. High rates of tactile hyperactivity, such as pain insensitivity, are also consistent with findings from a Shank3 animal model [28].

Results from 12-week stability data suggest that the SAND may be a useful clinical outcome assessment, with good stability across scales. In addition, hyporeactivity may represent a novel target for treatment in clinical trials within the syndrome. Clinically, practitioners should be aware of safety concerns resulting from sensory hyporeactivity (e.g., under-responsiveness to warning sounds such as sirens; high pain/temperature thresholds; delayed or absent response to a car passing by). The results suggest that interventions targeting sensory processing are warranted, with a particular focus on global hyporeactivity.

For the first time, we also explored genotype–phenotype associations in sensory reactivity symptomotology. Overall, results suggest that haploinsufficiency of *SHANK3* is the main driver of the sensory phenotype seen in PMS. These results are in line with previous genotype–phenotype studies where no differences were found in sensory processing as measured by the Short Sensory Profile [29] and the ADI-R sensory interests’ subdomain [11]. However, given the size of our cohort, these results should be interpreted with caution. Future studies should explore auditory processing in PMS in greater depth to determine whether the genotype–phenotype differences in auditory hyperreactivity and hyporeactivity persist in larger samples, which would indicate a greater level of hyporeactivity, and less hyperreactivity, in those with larger deletions. Investigating whether there may be other genes in the 22q13.3 region playing a role in auditory processing would be a meaningful future direction.

## 5. Conclusions

Using a comprehensive direct assessment of sensory symptomatology, this study established a distinct sensory reactivity phenotype in children with a PMS, characterized by global hyporeactivity across visual, tactile, and auditory modalities, and to a lesser degree, tactile and auditory sensory seeking. This sensory phenotype is distinct from iASD and other monogenic forms of ASD [21,30] and offers insight into the potential role of SHANK3 deficiency in sensory reactivity. Understanding sensory reactivity abnormalities in PMS in the context of known glutamatergic dysregulation may inform pharmacological treatment approaches.

## Figures and Tables

**Figure 1 genes-12-00977-f001:**
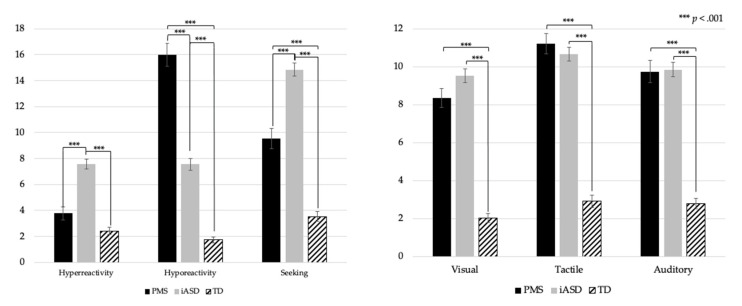
SAND domain (Hyperreactivity, Hyporeactivity, Seeking) and modality (Visual, Tactile, Auditory) mean scores for PMS, iASD, and TD groups.

**Table 1 genes-12-00977-t001:** Participant demographics. ADOS-2 comparison scores range from 1–10 with higher numbers reflecting greater symptom severity.

M(SD)	Male/Female	Age (years)	ASD Dx	ADOS-2 Comparison Score	Full Scale IQ/DQ	Nonverbal IQ/DQ	Verbal IQ/DQ	VABS Adaptive Behavior Composite
PMS	26/26	6.65 (2.90)	42/52	6.63 (2.23)	30.20 (16.88)	35.74 (19.66)	27.52 (19.53)	55.14 (12.61)
iASD	114/18	6.11 (2.55)	132/132	7.31 (1.60)	72.80 (29.99)	79.27 (30.17)	69.76 (30.66)	70.70 (15.39)
TD	24/30	5.39 (2.55)	0/54	N/A	117.21 (16.06)	115.18 (17.70)	114.88 (12.84)	N/A

IQ and Vineland Adaptive Behavior Composite scores are standard scores (*M* = 100; *SD* = 15). Developmental Quotients (DQs) were calculated by dividing age equivalents by chronological age x 100 for participants above the age-normed range on the Mullen Scales of Early Learning who were unable to complete the Stanford-Binet. Abbreviations: ADOS-2: Autism Diagnostic Observation Schedule (Second Edition); IQ: Intellectual quotient; DQ: Developmental quotient; VABS: Vineland Adaptive Behavior Scales; PMS: Phelan–McDermid syndrome; iASD: idiopathic autism spectrum disorder; TD: typically developing controls; N/A: not applicable.

**Table 2 genes-12-00977-t002:** Differences in SAND scores between groups. F and *p* values represent results from the MANOVA.

SAND Subscale	Group	Mean	SD	F	*p*
Hyperreactivity Total	TD	2.43	1.90	45.86	<0.001
	iASD	7.64	4.51		
	PMS	3.54	3.19		
Visual	TD	0.24	0.64	10.75	<0.001
	iASD	1.39	1.92		
	PMS	0.77	1.32		
Tactile	TD	1.06	1.34	16.70	<0.001
	iASD	2.74	2.17		
	PMS	1.60	1.87		
Auditory	TD	1.13	1.36	40.37	<0.001
	iASD	3.51	2.29		
	PMS	1.17	1.81		
Hyporeactivity Total	TD	1.74	1.65	114.89	<0.001
	iASD	7.55	5.23		
	PMS	16.19	6.32		
Visual	TD	0.74	1.05	74.45	<0.001
	iASD	3.05	2.28		
	PMS	5.63	2.28		
Tactile	TD	0.41	0.81	71.67	<0.001
	iASD	2.45	2.12		
	PMS	4.94	2.31		
Auditory	TD	0.59	0.92	68.50	<0.001
	iASD	2.05	2.45		
	PMS	5.62	2.85		
Seeking Total	TD	3.54	2.88	89.07	<0.001
	iASD	14.80	5.83		
	PMS	9.56	5.81		
Visual	TD	1.04	1.39	85.91	<0.001
	iASD	5.04	2.39		
	PMS	1.90	2.00		
Tactile	TD	1.44	1.69	47.79	<0.001
	iASD	5.44	2.75		
	PMS	4.67	2.72		
Auditory	TD	1.06	1.38	31.75	<0.001
	iASD	4.32	2.85		
	PMS	2.98	2.68		

**Table 3 genes-12-00977-t003:** Percentage of participants falling in the clinically significant range on the SAND.

	Total	Hyperreactivity	Hyporeactivity	Seeking	Visual	Tactile	Auditory
PMS	92.31%	11.54%	92.31%	65.38%	57.69%	80.77%	69.23%
iASD	93.18%	44.70%	43.18%	87.88%	65.15%	80.30%	71.97%

**Table 4 genes-12-00977-t004:** Stability of SAND scores over 12 weeks.

	Dx	ICC	95% CI Range	Sig
SAND				
Total	PMS	0.879	0.674–0.956	<0.001
	iASD	0.834	0.653–0.920	<0.001
Hyperreactivity	PMS	0.736	0.296–0.903	0.003
	iASD	0.665	0.297–0.839	0.002
Hyporeactivity	PMS	0.775	0.368–0.919	0.003
	iASD	0.766	0.512–0.887	<0.001
Seeking	PMS	0.821	0.495–0.936	<0.001
	iASD	0.91	0.813–0.957	<0.001
Visual	PMS	0.857	0.617–0.948	<0.001
	iASD	0.617	0.196–0.817	0.006
Auditory	PMS	0.927	0.804–0.973	<0.001
	iASD	0.766	0.513–0.888	<0.001
Tactile	PMS	0.615	−0.109–0.863	0.038
	iASD	0.837	0.666–0.921	<0.001

**Table 5 genes-12-00977-t005:** Genotype–phenotype associations. Mean and standard deviation displayed for SAND scores followed by proportion of participants who surpassed the clinically significant threshold on the SAND for each domain and subscale. * indicates a statistically significant difference.

	Scores			Proportion Surpassed Clinical Threshold		
Domain/Subscale	Class I	Class II	*p*	Class I	Class II	*p*
Hyperreactivity	3.83 (3.6)	3.14 (2.5)	0.679	16.67%	4.55%	0.176
Hyporeactivity	14.73 (6.2)	18.18 (6.0)	0.086	86.67%	100.00%	0.075
Seeking	10.07 (5.4)	8.86 (6.4)	0.504	70.00%	59.09%	0.414
Visual	7.83 (4.2)	8.95 (2.8)	0.199	46.67%	72.73%	0.06
Auditory	10.93 (3.7)	11.59 (4.3)	0.418	80.00%	81.82%	0.869
Tactile	9.87 (4.9)	9.64 (3.3)	0.918	66.67%	72.73%	0.64
Visual Hyperreactivity	0.80 (1.3)	0.73 (1.4)	0.798	0.00%	4.55%	0.238
Visual Hyporeactivity	5.10 (2.4)	6.36 (1.9)	0.073	66.67%	77.27%	0.404
Visual Seeking	1.93 (2.1)	1.86 (1.9)	0.977	16.67%	9.09%	0.429
Tactile Hyperreactivity	1.27 (1.6)	2.05 (2.1)	0.178	3.33%	4.55%	0.822
Tactile Hyporeactivity	4.83 (1.9)	5.09 (2.8)	0.562	66.67%	59.09%	0.575
Tactile Seeking	4.83 (2.3)	4.45 (3.3)	0.702	60.00%	45.45%	0.299
Auditory Hyperreactivity	1.77 (2.1)	0.36 (0.8)	0.011 *	16.67%	0.00%	0.044 *
Auditory Hyporeactivity	4.80 (3.0)	6.73 (2.3)	0.02 *	53.33%	81.82%	0.033 *
Auditory Seeking	3.30 (2.6)	2.55 (2.8)	0.26	26.67%	27.27%	0.961
Total	28.63 (11.1)	30.18 (7.5)	0.364	90.00%	100.00%	0.075

## Data Availability

The majority of the dataset used during the current study is included in this published article. The remainder of the dataset is available from the corresponding author on reasonable request and may require ethics review.

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
