# Peer review of "Sensory Reactivity Phenotype in Phelan–McDermid Syndrome Is Distinct from Idiopathic ASD"

_genes, 2021, doi:10.3390/genes12070977_

Round 1

Reviewer 1 Report

Tavassoli et al., investigated the sensory reactivity phenotype in the patients with PMS and found the difference between idiopathic ASD patients. This is a very interesting and comprehensive human study. I have two questions.

  1. Gender discrepancy is a hot topic in the ASD. Except the Genotype-phenotype associations in the Table 5, I am curious whether there is any gender-phenotype associations which contribute to the sensory reactivity difference. If not, would you please add several sentences in the discussion.
  2. Shank3 is a high-risk factor for ASD, which is well known from PMS. There are some sensory related behaviors determined by using Shank 3-deficiency mouse models, such as PPI. Would you add some results in the discussion?

Author Response

  1. Gender discrepancy is a hot topic in the ASD. Except the Genotype-phenotype associations in the Table 5, I am curious whether there is any gender-phenotype associations which contribute to the sensory reactivity difference. If not, would you please add several sentences in the discussion.

Thank you for your review and for this helpful suggestion for an additional analysis of gender. There were no gender-phenotype associations. This is now stated in the results and discussion.  

  1. Shank3 is a high-risk factor for ASD, which is well known from PMS. There are some sensory related behaviors determined by using Shank 3-deficiency mouse models, such as PPI. Would you add some results in the discussion?

We now include a relevant article showing differences in pain perception in an animal model of Shank3.

Reviewer 2 Report

The manuscript under review examined sensory reactivity phenotype and genotype-phenotype interactions in a sample of children with ASD. The authors examined sensory reactivity in a group of children with PMS compared with typically developing children and children with idiopathic autism spectrum disorder (iASD). Moreover, the authors split PMS into Class I deletions or sequence variants and larger Class II deletions, to further examine genotype-phenotype relationships. The authors conclude that haploinsufficiency of SHANK3 is the main driver of the sensory phenotype seen in PMS. I find the study very relevant to the field as it delineates the sensory reactivity phenotype in PMS by:  1) using a control group of typically developing children and a group of children with idiopathic ASD; 2) using a measure of sensory reactivity that includes direct observation and parent report and 3) using a genotype- phenotype approach. I think the study is well designed and have only a few suggestions/ comments that I would like the authors to address.  

Comments/ suggestions

  1. The authors end paragraph 1 of Introduction by describing sensory reactivity within the context of DSM-5 (lines 55-58). I think the concept of sensory reactivity could be extended by describing each of the patterns (hyper, hyper and sensory seeking), the modalities (visual, auditory and tactile) and by providing examples of everyday behaviors that result from abnormal/ atypical sensory processing.
  2. Similarly, I think it would be important to include a more detailed background on sensory reactivity symptoms in ASD (line 59). This is useful to guide the reader on the differences in phenotypes between idiopathic ASD and individuals with PMS. This information should also be cited in the discussion section. Some of the work that could be cited include for example Baranek et al., Cascio et al., or Green et al. (not limited to).
  3. On Table 1 it is not clear what each variable means - please include the labels for each variable (FSIQ/DQ; NVIQ/DQ; etc.) under caption.
  4. Under Methods Section - Sensory Evaluation (line 129) I think it would be useful to include a couple of examples on the type of stimuli presented and hyper/ hypo/ seeking behaviors for a given stimuli.
  5. Under Results Section it is mentioned that Children with PMS and iASD significantly differed on all 3 domains, but not on any modality. If the domains are created by items from the 3 modalities, how do you explain this result?
  6. Although the Genotype-Phenotype associations are very interesting, they should be interpreted with caution give the sample size of 30 and 22 participants in each group. The authors report Cohen’s d of .72 and .89 but these tests are not mentioned in the Analysis Section. The authors should report the statistical approach under the Analysis Section and state if the study is powered to conduct these comparisons. In addition, I encourage the authors to include sample size as a limitation under discussion section.
  7. Under the Discussion, line 259 it is mentioned that “interventions targeting sensory processing are warranted, focusing on global hyporeactivity and sensory seeking” – it is not clear to me why sensory seeking is included here, please clarify.
  8. I think the authors could expand a little on the differences found between iASD and PMS syndrome sensory reactivity phenotypes, and how this line of research can be useful to inform about both conditions and what the next steps are. A section listing study limitations and future directions is encouraged.

Author Response

  1. The authors end paragraph 1 of Introduction by describing sensory reactivity within the context of DSM-5 (lines 55-58). I think the concept of sensory reactivity could be extended by describing each of the patterns (hyper, hyper and sensory seeking), the modalities (visual, auditory and tactile) and by providing examples of everyday behaviors that result from abnormal/ atypical sensory processing.

Thank you for pointing this out, we now describe each sensory reactivity difference in greater depth; for example, “Hyperreactivity describes a strong and/or aversive reaction to a sensory stimulus, such as covering ears in response to every day sounds or avoiding wearing clothes of certain texture. Hyporeactivity describes delayed or absent responses to sensory stimuli such as failure to notice the sound of an alarm or the sight of a car passing by. Sensory seeking refers to a craving and fascination with certain sensory stimuli, such as repeatedly touching specific textures or visually inspecting objects for extended periods of time.”

  1. Similarly, I think it would be important to include a more detailed background on sensory reactivity symptoms in ASD (line 59). This is useful to guide the reader on the differences in phenotypes between idiopathic ASD and individuals with PMS. This information should also be cited in the discussion section. Some of the work that could be cited include for example Baranek et al., Cascio et al., or Green et al. (not limited to).

We have now added additional text and references, however, given the word limit we cannot extensively describe sensory reactivity differences in idiopathic forms of autism.

  1. On Table 1 it is not clear what each variable means - please include the labels for each variable (FSIQ/DQ; NVIQ/DQ; etc.) under caption.

We added a list of abbreviations after the table caption.  We also removed certain abbreviations in the table for easier review. 

  1. Under Methods Section - Sensory Evaluation (line 129) I think it would be useful to include a couple of examples on the type of stimuli presented and hyper/ hypo/ seeking behaviors for a given stimuli.

Thank you for this suggestion. We now include examples of stimuli presented, please see: “Children are presented with a series of visual (e.g., spinning disc, light up wand), tactile (e.g., textured balls, hot/cold packs) and auditory stimuli (e.g., unexpected noisemaker, siren buzzer)…”Importantly, there are not toys specifically for hypereactivity, hyporeactivity or seeking, but rather behaviors are observed and scored throughout the observation. For example, one child may seek out a textured ball (e.g., pressing to skin repeatedly), while another child may avoid touching it.

  1. Under Results Section it is mentioned that Children with PMS and iASD significantly differed on all 3 domains, but not on any modality. If the domains are created by items from the 3 modalities, how do you explain this result?

Modalities are a combination of hyperreactivity, hyporeactivity and seeking. As children with PMS have more hyporeactivity symptoms and less hyperreactivity symptoms these will cancel each other out when only looking at modalities.

  1. Although the Genotype-Phenotype associations are very interesting, they should be interpreted with caution give the sample size of 30 and 22 participants in each group. The authors report Cohen’s d of .72 and .89 but these tests are not mentioned in the Analysis Section. The authors should report the statistical approach under the Analysis Section and state if the study is powered to conduct these comparisons. In addition, I encourage the authors to include sample size as a limitation under discussion section.

We agree sample size is a limitation and have now included this in the Discussion.  We also added a sentence in the analysis section about effect sizes. 

  1. Under the Discussion, line 259 it is mentioned that “interventions targeting sensory processing are warranted, focusing on global hyporeactivity and sensory seeking” – it is not clear to me why sensory seeking is included here, please clarify.

Thank you for pointing this out. We have now corrected this sentence by removing mention of sensory seeking.

  1. I think the authors could expand a little on the differences found between iASD and PMS syndrome sensory reactivity phenotypes, and how this line of research can be useful to inform about both conditions and what the next steps are. A section listing study limitations and future directions is encouraged.

Thank you for this comment, please see the following in the revised manuscript: “However, given the size of our cohort, these results should be interpreted with caution. Future studies should explore auditory processing in PMS in greater depth to determine whether the genotype-phenotype differences in auditory hyperreactivity and hyporeactivity persist in larger samples, which would indicate a greater level of hyporeactivity, and less hyperreactivty, in those with larger deletions. Investigating whether there may be other genes in the 22q13.3 region playing a role in auditory processing would be a meaningful future direction.”